# Selectively tunable optical Stark effect of anisotropic excitons in atomically thin ReS$_2$

Sangwan Sim[1,*], Doeon Lee[1,*], Minji Noh[1], Soonyoung Cha[1], Chan Ho Soh[1], Ji Ho Sung[2,3], Moon-Ho Jo[2,3,4] & Hyunyong Choi[1]

The optical Stark effect is a coherent light–matter interaction describing the modification of quantum states by non-resonant light illumination in atoms, solids and nanostructures. Researchers have strived to utilize this effect to control exciton states, aiming to realize ultra-high-speed optical switches and modulators. However, most studies have focused on the optical Stark effect of only the lowest exciton state due to lack of energy selectivity, resulting in low degree-of-freedom devices. Here, by applying a linearly polarized laser pulse to few-layer ReS$_2$, where reduced symmetry leads to strong in-plane anisotropy of excitons, we control the optical Stark shift of two energetically separated exciton states. Especially, we selectively tune the Stark effect of an individual state with varying light polarization. This is possible because each state has a completely distinct dependence on light polarization due to different excitonic transition dipole moments. Our finding provides a methodology for energy-selective control of exciton states.

[1] School of Electrical and Electronic Engineering, Yonsei University, Seoul 120-749, Korea. [2] Center for Artificial Low Dimensional Electronic Systems, Institute for Basic Science (IBS), Pohang University of Science and Technology (POSTECH), 77 Cheongam-Ro, Pohang 790-784, Korea. [3] Division of Advanced Materials Science, Pohang University of Science and Technology (POSTECH), 77 Cheongam-Ro, Pohang 790-784, Korea. [4] Department of Materials Science and Engineering, Pohang University of Science and Technology (POSTECH), 77 Cheongam-Ro, Pohang 790-784, Korea. * These authors contributed equally to this work. Correspondence and requests for materials should be addressed to H.C. (email: hychoi@yonsei.ac.kr).

When a semiconducting system is excited by a laser pulse with photon energy lower than that of exciton transition, a virtual optical transition is invoked resulting in so-called photon-dressed states[1–16]. It usually interacts repulsively with original states, leading to the characteristic blue-shift of the exciton energy spectrum. Along with the fact that the coherent interaction can only take place during the time duration of an ultra-short laser pulse, such a unique feature makes this phenomena, the so-called excitonic optical Stark effect, ideal for ultrafast optical switches and modulators[3–10,12–18]. So far, however, there has been no strategy for energy-selective control of exciton states. More specifically, in conventional semiconductors such as GaAs-based quantum wells, most relevant studies have been focused only on the lowest exciton state (for example, heavy-hole exciton), because it was impossible to selectively measure the optical Stark shift of the higher state (for example, light-hole exciton), as illustrated in Fig. 1a (refs 4,5,9,11,12). Thus, this effect still lacks practicality, with possible applications such as wavelength-selective optoelectronics. In this regard, if it is possible to selectively control the Stark shift of more than one exciton state, it shall be a technological breakthrough for novel optical devices with high degree-of-freedom and functionality.

In the past few years, two-dimensional transition metal dichalcogenides (2D TMDs) have gained intensive attention due to their outstanding excitonic properties, arising from strong confinement and reduced dielectric screening[19–21]. It has recently been discovered that the optical Stark shift of excitons in different valleys in momentum space can be determined by changing helicity of light in monolayer group VI TMDs ($WS_2$ and $WSe_2$) in a completely selective manner[13,14]. Since the valley excitons at K (K') point in these studies are energetically indistinguishable, however, it can be said that no experimental approaches for energy-selective optical Stark effect of excitons have been made.

$ReS_2$ is a member of a recently emerged family of group VII 2D TMDs[22–29]. Unlike molybdenum and tungsten dichalcogenides, group VI TMDs with hexagonal structure, $ReS_2$ exhibits reduced in-plane crystal symmetry with a distorted 1T structure forming Re atom chains (bluish green dots in Fig. 1c) aligned along the $b$ axis (yellow thick line in Fig. 1c)[23]. Remarkably, the unique symmetry leads to the anisotropic linear polarization of excitons, as illustrated by the blue and red electron–hole pairs in Fig. 1c. Figure 1d shows the polarization ($\theta$)-resolved absorption spectra of few-layer (7–8 layers) $ReS_2$ with linearly polarized light (where $\theta$ measures the polarization angle with respect to the $b$ axis, see the inset of Fig. 1c). The absorption peaks, labeled as $X_1$ and $X_2$, arise from the two lowest, energetically nondegenerate direct exciton states near $\Gamma$ point[27] and show strong polarization dependence as reported by Aslan et al.[28] Corresponding $\theta$-dependent Lorentzian spectral weights clearly reveal their linear nature (Fig. 1c). Importantly, these excitons are polarized at different angles ($\theta \sim 19°$ and $\sim 87°$ for $X_1$ and $X_2$, respectively)[28],

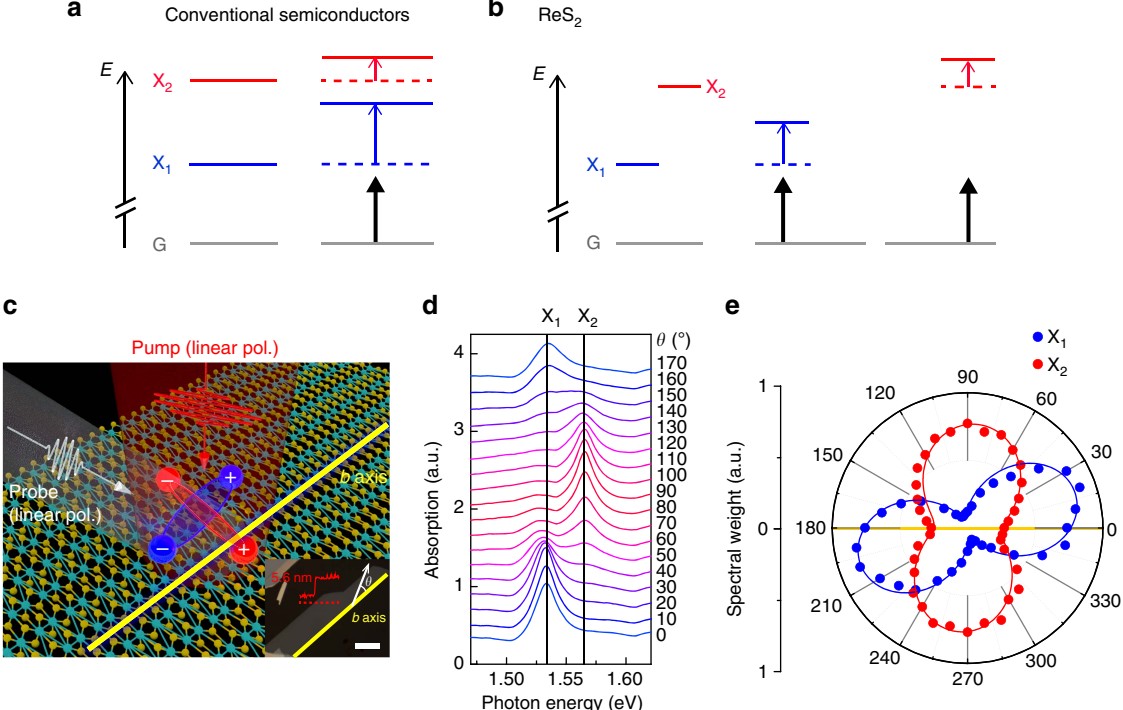

**Figure 1 | Light-polarization-dependent optical Stark effect of excitons in ReS$_2$.** (**a,b**) Comparison of excitonic optical Stark effect in conventional semiconductors (**a**) and ReS$_2$ (**b**). $X_1$ and $X_2$ are the lowest two exciton states and G indicates the ground state. In conventional semiconductors (**a**), such as GaAs-based quantum wells, $X_1$ and $X_2$ correspond to heavy-hole and light-hole excitons, respectively. Pump photon energy should be smaller than that of the lower-lying $X_1$ transition to avoid real transition (see black arrows). Dashed lines represent the energy levels of unperturbed states. Unlike conventional semiconductors, energy-selective optical Stark effect is possible in ReS$_2$ by choosing proper polarization configurations of pump and probe light (see main text). (**c**) A schematic illustrating the pump-probe experiment of few-layer ReS$_2$. Bluish-green (yellow) dots are Re (S) atoms. Two electron ( − )—hole ( + ) pairs represent anisotropic excitons, $X_1$ (blue) and $X_2$ (red). The light-polarization-dependent optical Stark effect is measured by varying polarization of the linearly polarized pump (red field) and probe (white field) pulses. Inset: optical image of few-layer ReS$_2$. Yellow thick lines (main panel and inset) indicate the crystal $b$ axis. Red graph shows the AFM height profile of the ReS$_2$ sample along the red dashed line. Scale bar, 15 μm. (**d**) Polarization-dependent absorption spectra of few-layer ReS$_2$. (**e**) Corresponding spectral weights of Lorentzian contributions of $X_1$ (blue dots) and $X_2$ (red dots). Yellow line represents the $b$-axis. Solid fit lines are proportional to $\cos^2(\theta - \theta_{max})$ with offset, where $\theta_{max}$ values are polarization angles of the excitons (19° for $X_1$ and 87° for $X_2$).

making it possible to access individual exciton states selectively by choosing the appropriate polarization of light. In view of the excitonic optical Stark effect, therefore, it can be expected that these two exciton states will be selectively controlled by tuning light polarization as depicted in Fig. 1b; the optical Stark shift will predominantly occur at the $X_1$ ($X_2$) state when the polarizations of pump and probe light are parallel with its orientation angle of $\sim 19°$ ($\sim 87°$), as shown in the middle (right) panel of Fig. 1b.

In this work, by using ultrafast optical pump-probe spectroscopy, we control the optical Stark effect of the two direct exciton states in a light-polarization-selective manner. Based on the excitons' linearly polarized nature and anisotropy, we selectively measure the shift of the two energetically nondegenerate states by manipulating the angle of the linearly polarized pump and probe light. We gradually tune the Stark shift for $X_1$ and $X_2$, which obviously has different light-polarization dependence. This is possible because the transition dipole moment of each excitonic state depends on completely different angles of light polarization. Our findings offer a foundation for energy-selective control of quantum states in excitonic systems.

## Results

**Observation of the optical Stark effect in ReS$_2$.** Few-layer (7–8 layers) ReS$_2$ flakes were mechanically exfoliated onto sapphire substrates as shown in the inset of Fig. 1c (see Method). We chose to study few-layer ReS$_2$ over monolayer due to the following reasons: the difference between the polarization angle of $X_1$ and $X_2$ is larger in the few-layer ($\sim 70°$) than in the monolayer ($\sim 45°$), and the relative oscillator strength of the $X_1$ transition is extremely small compared with the $X_2$ transition in the few-layer[28]. These features indicate that the monolayer is less favourable in experimentally distinguishing the optical response of the individual excitons.

We measured the pump-induced change in the transmission of probe light (differential transmission, DT) as a function of pump–probe time delay ($\tau$) in pump–probe experiments (see Methods). Polarization-controlled pump and probe beams were both linearly polarized. All measurements were performed at low temperature (78 K) because exciton linewidths in ReS$_2$ are significantly narrowed with decreasing temperature[22] (see Supplementary Figs 1–3 and Supplementary Note 1 for room temperature measurements of bilayer ReS$_2$ with circularly polarized probe).

We first explore the detailed DT response with co-linear pump–probe configuration at $\theta = 70°$ to confirm the optical Stark effect of both $X_1$ and $X_2$ excitons. Two absorption peaks due to $X_1$ and $X_2$ are observed near 1.53 and 1.59 eV, respectively, which is in well agreement with a prior study[28] (Fig. 2a). To measure the optical Stark shifts of these two states, we excited the sample with pump photon energy detuned to 90 meV below $X_1$ transition (that is, pump photon energy = 1.44 eV) and monitored the time-resolved DT dynamics. Before analyzing the results, it is instructive to note the spectral signature of the optical Stark effect. As illustrated in Fig. 2b, when an absorption resonance is blue-shifted, the corresponding DT signal shows a positive-to-negative sign change near the resonance energy, resulting in a similar shape to the first derivative of the absorption. Indeed, we observed positive-to-negative sign changes of DT near $X_1$ and $X_2$ at $\tau = 0$ fs (Fig. 2c), indicating blue shifts of both exciton resonances. We also see that the transient shifts of excitons occur only during the pump laser time duration. Such a fast response cannot arise from the slow dynamics of photo-generated excitons. Instead, it stems from the coherent interaction of the material with ultra-short pulse, namely the excitonic optical Stark effect. This is corroborated by the DT time-traces (Fig. 2d) which show strong spike-like signals due to transient shifts of excitons near

$\tau = 0$ fs (see Supplementary Fig. 4 and Supplementary Note 2 for further confirmation). The spike-like peaks are followed by slow DT signals arising from pump-excited real carrier dynamics[5,6,13]. We eliminate this effect when estimating the magnitude of Stark shifts in the discussion below (see Supplementary Fig. 5 and Supplementary Note 3).

**Energy-selective optical Stark effect.** With this understanding, we now investigate the selectively tunable optical Stark effect of excitons. For this purpose, we measure the DT spectra at $\tau = 0$ fs while varying pump–probe polarization configuration, with pump photon energy detuned to 90 meV below the $X_1$ level and fluence fixed at 230 µJ cm$^{-2}$. First, to measure the Stark shift of $X_1$ in a selective manner, the probe polarization angle was fixed at $\theta = 20°$, at which $X_1$ is predominantly coupled with light and $X_2$ has a negligible transition dipole moment (see the equilibrium absorption spectrum in the top panel of Fig. 3a). Under this condition, we observed that the co-linearly polarized pump–probe pulses cause an absorption-derivative-like DT response (middle panel, Fig. 3a) only at the spectral region dominated by $X_1$ (blue-shaded area), indicating selective optical Stark effect of $X_1$. The DT spectrum has slight asymmetry and broad background, originating from the pump-excited real carriers discussed above (Supplementary Note 3). The amplitude of the Stark signal becomes small when the pump is orthogonally polarized to the probe, as shown in the bottom panel of Fig. 3a, showing reduced blue shift. This can be explained by the anisotropic optical selection rule of $X_1$, which show very weak coupling with pump light with the polarization of $\theta = 110°$ (Fig. 1e). In a similar manner, we selectively measure the optical Stark shift of $X_2$. At a fixed probe polarization of $\theta = 90°$, at which $X_2$ dominates the optical response (top panel in Fig. 3b), clear absorption-derivative-like DT response due to optical Stark shift of $X_2$ is observed at co-polarized pump–probe configuration (middle panel in Fig. 3b). Similar to the $X_1$'s response, it shows decrease in amplitude at the cross-polarized pump–probe configuration (bottom panel in Fig. 3b).

These results enlighten us of significant benefits of ReS$_2$ in terms of selective optical control of excitons. Firstly, as shown in the middle panels of Fig. 3a,b, it is possible to measure the shift of a certain exciton state in a completely exclusive manner, indicating high exciton selectivity. More importantly, the results also reveal energy selectivity, considering that the two exciton states possess well-separated energy levels (note that the spectral distance between the two exciton resonances are larger than the sum of their half linewidths, see Supplementary Fig. 6). In particular, the higher-lying exciton state ($X_2$) can be selectively modulated without being disturbed by the lower-lying exciton ($X_1$) (Fig. 3b). Such unique functionality is absent in other materials, such as semiconductor quantum wells, carbon nanotubes and group VI TMDs. Schematics in Fig. 1b summarize these findings.

For a more comprehensive understanding of this effect, we measured the pump-polarization-dependent DT spectra while tuning the angle of pump polarization continuously at $\tau = 0$ fs, and estimated corresponding magnitudes of exciton resonance shifts for $X_1$ ($\Delta E_1$, blue dots in Fig. 3c) and $X_2$ ($\Delta E_2$, red dots in Fig. 3d) with fixed probe polarizations of $\theta = 20°$ and of $\theta = 90°$, respectively (see Supplementary Fig. 7 for measured DT spectra and Supplementary Note 3 for details of the fitting procedure). The Stark shift is maximized (minimized) when pump polarization is parallel (perpendicular) to the orientation angle of each exciton. As indicated by the solid lines in Fig. 3c,d, the gradual changes of $\Delta E_i$ ($i = 1, 2$) as a function of pump polarization ($\theta_{pump}$) can be well fit with a simple function $\Delta E_i(\theta_{pump}) = a + b \cos^2(\theta_{pump} - \theta_{max,i})$, setting $\theta_{max,i}$ (at which

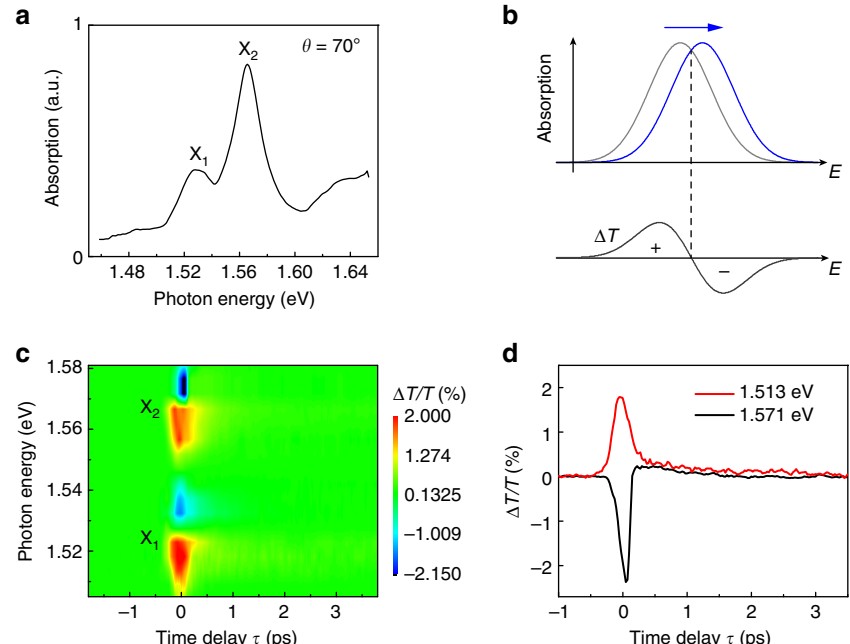

**Figure 2 | Observation of the optical Stark effect in few-layer ReS$_2$.** (**a**) Absorption spectrum with light polarization angle of $\theta = 70°$. (**b**) Schematic illustration of DT spectrum (black line) due to Stark shift for an exciton state. Gray and blue lines indicate unperturbed and blue-shifted absorption resonances, respectively. (**c,d**) Transient DT dynamics with a co-polarized pump–probe configuration ($\theta = 70°$) (**c**) and time traces at two different probe photon energies (**d**).

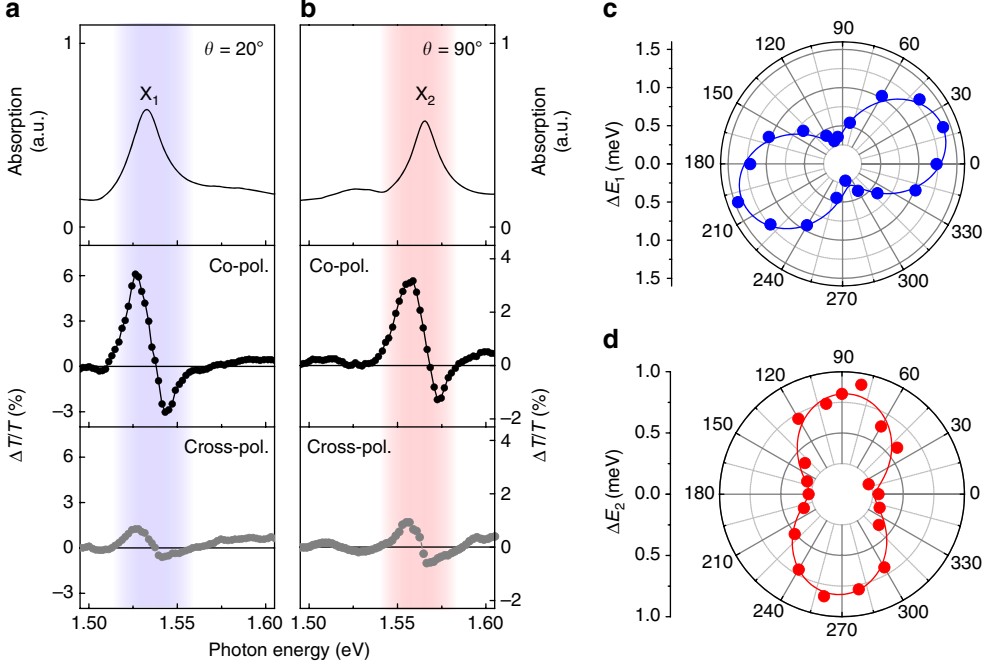

**Figure 3 | Exciton-selective optical Stark effect controlled by light polarization.** (**a,b**) Equilibrium absorption spectra with light polarization angle of $\theta = 20°$ (top panel in **a**) and $\theta = 90°$ (top panel in **b**). Black dots (middle panels) and gray dots (bottom panels) are transient DT spectra at $\tau = 0$ fs with co-polarized and cross-polarized pump-probe configurations, respectively. Probe polarization angles are fixed at $\theta = 20°$ (**a**) and $\theta = 90°$ (**b**). Pump photon energy is 1.44 eV with fluence of 230 μJ cm$^{-2}$. The blue- (red-) shaded area represents the spectral region where DT signal is dominated by the shift of the X$_1$ (X$_2$) state. (**c,d**) Pump-polarization-dependent optical Stark shifts of X$_1$ (blue dots in **c**) and X$_2$ (red dots in **d**) with fixed probe polarizations of $\theta = 20°$ and $\theta = 90°$, respectively. The solid lines are fits.

$\Delta E_i(\theta_{pump})$ is maximized) to the individual excitons' original polarizations (that is, $\theta_{max,1} = 19°$ for X$_1$ and $\theta_{max,2} = 87°$ for X$_2$). Here, $a$ and $b$ are the fitting parameters. These results clearly indicate that the polarization-dependent optical Stark shifts of each exciton states directly follow their own spatial orientations.

Such behavior can be understood by the following simple relation. In the semi-classical picture, the optical Stark shift of an individual exciton state can be approximated as $\Delta E = \mu^2 \varepsilon^2 / (\hbar\omega_X - \hbar\omega_{pump})$, where $\mu$ is the transition dipole moment between the ground and the exciton state, $\varepsilon$ is the electric

field of the pump light, $\hbar\omega_X$ and $\hbar\omega_{pump}$ are the energy of the excitonic transition and pump light, respectively. Considering the proportionality of $\mu^2$ to the oscillator strength, the observed $\theta_{pump}$ dependence of $\Delta E_i$ is well explained by the light-polarization-dependent spectral-weights (see Fig. 1e) for each excitonic absorption. Note that we obtained similar results with a circularly polarized probe in bilayer $ReS_2$ (Supplementary Note 1). The strengths of the optical Stark effect ($S = \Delta E_i \times (\hbar\omega_X - \hbar\omega_{pump})/2\varepsilon^2$)[12,14] for $X_1$ and $X_2$ are about $\sim 17$ $D^2$ and $\sim 15$ $D^2$ at co-linear pump–probe polarization configurations, respectively ($1D \approx 3.3 \times 10^{-30}$ C·m). These values are of the same order of magnitude as that of group VI TMD ($\sim 45$ $D^2$; ref. 14).

## Discussion

This work reveals functionalities of group VII 2D TMD $ReS_2$ for ultrafast optical applications. By utilizing the characteristic anisotropy of excitons, we can selectively tune the optical Stark shift of two energetically nondegenerate exciton states by manipulating light polarization. We emphasize that such advantages basically originate from the unique in-plane crystal anisotropy of group VII TMDs, which is absent in group VI 2D TMDs (for example, $MoS_2$, $MoSe_2$, $WS_2$ and $WSe_2$). Of course, group VII TMDs are not the only material family exhibiting anisotropic property of excitons; there are several systems possessing anisotropic excitonic properties (such as carbon nanotubes (CNTs) and black phosphorus (BP))[12,30–32]. However, both of them lack polarization-dependent exciton selectivity so that energy-selective optical Stark effect cannot be expected. For CNTs, since the anisotropy of excitonic transition arises simply from the geometrical alignment, all excitonic transitions should have same polarization dependence[12]. For BP, there is only one prominent excitonic transition with distinct anisotropy[30]. Thus, group VII TMDs are ideals material platforms for testing the energy selective control of the excitonic optical Stark effect.

## Methods

**Sample preparation.** The $ReS_2$ few-layer flake on a sapphire substrate was prepared by polydimethylsiloxane (PDMS)-assisted mechanical exfoliation from bulk crystals (HQ graphene).

**Ultrafast spectroscopy.** Ultrafast pump–probe spectroscopy was based on the 250 KHz Ti:sapphire regenerative amplifier laser system (Coherent RegA 9050). Pump pulses of 860 nm with filtered 10 nm (time duration $\sim 200$ fs) bandwidth were obtained by the second harmonic generation of an optical parametric amplifier's idler output pulses (coherent OPA). White-light-continuum probe pulses were generated by focusing 800 nm pulses to a sapphire disk. The linear polarizations of pump and probe pulses were controlled by sets of polarizers and half-wave plates. We measured pump-induced percent change in the transmission of probe light ($\Delta T/T$). All measurements were performed at 78 K.

**Data availability.** The data that support the findings of this study are available from the corresponding author upon reasonable request.

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

## Acknowledgements

S.S., D.L., M.N., S.C., C.H.S. and H.C. were supported by the National Research Foundation of Korea (NRF) through the government of Korea (MSIP) (Grant Nos NRF-2015R1A2A1A10052520, NRF-2016R1A4A1012929), Global Frontier Program (2014M3A6B3063709), the Yonsei University Yonsei-SNU Collaborative Research Fund of 2014, and the Yonsei University Future-leading Research Initiative of 2014. J.H.S. and M.-H.J. were supported by Institute for Basic Science (IBS), Korea under the Project Code (IBS-R014-G1-2016-a00).

## Author contributions

H.C. and S.S. conceived the experimental idea. S.S, D.L., M.N., S.C., and C.H.S. carried out the experiments and performed theoretical analysis. D.L. prepared and characterized the sample. S.S., D.L., S.C., J.H.S., M.-H.J and H.C. contributed to interpretation of the measured data. S.S., D.L., C.H.S. and H.C. wrote the manuscript with inputs from the other authors. All authors discussed the results and commented on the manuscript.

## Additional information

**Competing financial interests:** The authors declare no competing financial interests.

**Publisher's note**: 

