## [Peer Review File · Nature Communications]

Reviewers' comments:

Reviewer #1 (Remarks to the Author):

The authors report an observation of optical Stark effect in atomically thin ReS₂. Prominent two excitonic resonances in ReS₂ have distinct dipole selection rules which sensitively depend on linear polarization of light. Authors use polarization-resolved optical pump-probe spectroscopy to demonstrate that optical Stark effect can selectively induce resonance energy shift with the control of pump pulse polarization. The presentation of their manuscript is clear and their main conclusion of the work is sound and mostly justified. Although the claimed application would be a little speculative and far-stretched, this work can potentially lead to further interesting physics and applications. I would recommend for a publication in Nature Communication.

However, I suggest to check one thing before the publication. I am concerned about their explanation on residual energy shift even when pump polarization is completely orthogonal to the resonance (Figure 3c). Could this be simply because two exciton resonances share either the same conduction band or valence band which forms the exciton states? If this is the case, it will simply explain their observation without introducing coherent coupling.

Reviewer #2 (Remarks to the Author):

The authors report the observation of a polarization-dependent optical stark effect in ReS₂. This arises due to the anisotropic formation of excitons in this material. The observations were made using ultrafast optical pump-probe techniques, which is the same technique used in previous observations of the optical stark effect. The observation of a polarization dependent optical stark effect is new, the data and analysis is convincing, and the paper is well written. I think it is highly suitable for publication in Nature Communications.

I would only suggest adding some words on how this anisotropic optical stark effect affects things in the two valleys at K and K'. This would be useful information since the previous publications on this subject have focused on the valley selectivity of the effect in this material class.

Reviewer #3 (Remarks to the Author):

The authors performed an optical Stark effect experiment on bilayer ReS₂. They claim that two different states can exhibit relatively different energy shifts depending on the laser polarization. They show additional fluence and time dependence measurements to support the above conclusion.

I have read carefully the main text and the supplementary, and below is my review concerning (i) the novelty, (ii) the quality/clarity, and (iii) the impact of this work, which are the criteria to maintain the high-standard journal of Nature Communications.

Regarding the novelty of this work, it is natural to make a comparison with the existing related works on the optical Stark effect found in the literature: Note that the optical Stark effect is already known for many years in atoms and in solids (References 1-11). Selectively tunable optical Stark effect has also been shown in transition-metal dichalcogenides (TMDs, References 12-13) and in lead-halide perovskites (Science Advances, DOI: 10.1126/sciadv.1600477), where two different exciton states can exhibit different energy shifts depending on the laser polarization. This is basically the same phenomenon that is claimed by the present author, and simply using linear instead of circular light polarization into a slightly different material is not sufficient to claim this as a new finding. In addition to bilayer ReS₂, there are many other materials that exhibit anisotropic

electronic and optical properties such as carbon nanotubes and black phosphorus, from which anisotropic optical Stark effect is expected. In this perspective, the present manuscript is too similar with existing works, and it lacks the novelty required for Nature Communications.

More fundamentally, note that the two states investigated in this study are already different in energy by 50 meV. Hence, any attempt to further shift their relative energies by merely 1 meV has only little significance for fundamental science and applications. This situation is different from the selective energy shift in TMDs and perovskites above because the two states are originally identical in energy and protected by a certain symmetry. Hence, shifting their relative energies is of significant interest in fundamental science and applications, which is not the case for bilayer ReS₂.

Regarding the quality of this work, I think the measured DT spectra (Fig 2c, Fig 2e, Fig 3a) can still benefit from (i) improving the signal-to-noise ratio and from (ii) acquiring finer interval data points, as compared to the better data quality in the earlier works mentioned above. Besides, these DT spectra do not show a straightforward interpretation of an energy shift, where a simple derivative-like curve should be expected. This is because the two states are not well separated in energy, with energy separation that is comparable to their peak widths. I have no doubt that the optical contribution from the optical Stark effect does exist, as the authors have provided their best efforts to show it in their analysis. But again the compromising data quality makes it difficult to disentangle the contributions from possible coherent spectral oscillations (Phys. Rev. Lett. 59, 2588 (1987), Optics Letters 13, 276 (1988)) or from other long-lived dissipative processes. This also makes difficult to accurately determine the magnitude of the energy shift that, as of now, can be too sensitive to the input fitting parameters. I am afraid that the lack of clarity in the data may be confusing for some readers from interdisciplinary background.

The authors emphasize on the word "selectivity" in this work, but according to Figure 3c the selectivity is only up to a factor of 1.75-to-0.50. This is a rather poor contrast as compared to other existing works, and it is rather impractical for applications, in contrast to the authors' claim.

Regarding the impact of this work, "Science-wise" the anisotropic property of this material is already known and the polarization-selective optical Stark effect is already demonstrated in previous works, "Applications-wise" it is a little difficult to say because the observed effect (1 meV) is much smaller than the linewidth and the thermal energy at room temperature. I think currently it is rather important to study the equilibrium phenomena of this material more rigorously. For example, the equilibrium electronic structure of ReS₂ still suffers from controversial reports on whether the lowest energy gap corresponds to a direct or an indirect transition (References 18 and 23), or whether the interlayer coupling is really insignificant (References 18 and Nano Lett. 16, 1404 (2016) etc.). This controversy could affect the data interpretation of the present work. This is extremely important especially considering the high standard Nature Communications that maintains the novelty, the high data quality, and the correct interpretation.

Therefore, I cannot recommend the publication of this work in Nature Communications. But I think the authors can still consider submitting their works in a more specialized journals, possibly in the ACS or AIP journals. Also, it would be good to use less excessive phrase in the revised manuscript:

- The phrase "multiple energy levels" is mentioned several times (Line 30, 58, 82, 150), while only two energy levels are relevant in this work. This can be potentially confusing because the readers would expect a multiple number of energy levels like 5 or more.
- Line 46, 57, 82, "So far there has been no strategy for accessing multiple energy levels of excitons in a selective manner." However, selective tuning of two different excitons has been demonstrated in TMDs and perovskites above. Also, manipulating different energy levels can also be done simply by tuning the excitation photon energy.

In particular, the following sentences in the current manuscript are stretching too far into

pseudoapplications because the observed effect is too small:

- Line 37, 163, 169, "... we finally reveal a new applicability of ReS2 for modulating optical transmittance in the real-time domain." Note that the 1 meV shift is much smaller than the linewidth, and much smaller than the thermal energy at room temperature, thus rendering this effect impractical for the functionalities mentioned in the conclusions.

Point-by-point responses to the issues raised by the reviewers

General remarks of Reviewer 1:

The authors report an observation of optical Stark effect in atomically thin ReS₂. Prominent two excitonic resonances in ReS₂ have distinct dipole selection rules which sensitively depend on linear polarization of light. Authors use polarization-resolved optical pump-probe spectroscopy to demonstrate that optical Stark effect can selectively induce resonance energy shift with the control of pump pulse polarization. The presentation of their manuscript is clear and their main conclusion of the work is sound and mostly justified. Although the claimed application would be a little speculative and far-stretched, this work can potentially lead to further interesting physics and applications. I would recommend for a publication in Nature Communication.

Response:

We appreciate the time that Reviewer 1 took to read our paper. We are quite pleased that reviewer 1 found our work is sound and interesting. We are also very grateful for his/her thoughtful comments on the light-polarization dependence of the observed excitonic optical Stark shifts. This has been very helpful for us in improving the quality of our manuscript. Below we present our response to the reviewer 1's comments.

Comments 1-1:

However, I suggest to check one thing before the publication. I am concerned about their explanation on residual energy shift even when pump polarization is completely orthogonal to the resonance (Figure 3c). Could this be simply because two exciton resonances share either the same conduction band or valence band which forms the exciton states? If this is the case, it will simply explain their observation without introducing coherent coupling.

Response 1-1:

We thank the reviewer for her/his very important comments. In the measurement of the polarization-dependent optical Stark effect, we observed that the excitonic blue-shift takes place even when the polarization of the pump is perpendicular to the excitonic transition, as shown in Fig. 3c of the original manuscript. We attributed this to the coherent coupling of Stark shifts, based on the phenomenological similarity of our result to a prior study (Donovan *et al.*, Phys. Rev. Lett. 87, 237402 (2002)). However, we fully agree with Reviewer 1 that the shared conduction or valence band of the two excitonic transitions (labeled by X₁ and X₂)

explains the observed results much better, because such interpretation have been successfully figured out, showing similar results in many previous studies (e.g., Joffre *et al.*, Phys. Rev. Lett. 62, 74 (1988); Gupta *et al.*, Science 292, 2458 (2001); Choi *et al.*, Phys. Rev. B 65, 155206 (2002)). Moreover, introducing coherence generally requires more rigorous proof. For these reasons, we have revised our manuscript, as shown in Table. R1-1.

- Revised sentences in the main text:

Original	revised
(line 161) This may be due to the influence of the shift from one exciton state to the other via coherent coupling ¹⁰ .	(Supplementary Information S5) This may be because the two excitonic transitions share either the same conduction band or valence band which forms the exciton states ^{8,10,11} .

Table. R1-1. Revised sentence.

Regarding this issue, please note that we have inserted the revised sentence in Table. R1-1 into the section 5 of the Supplementary Information because we have transferred all original data to the Supplementary Information. In the revision process, we have re-measured full data sets with a thicker sample at low temperature (78 K) to improve the data quality (in the original manuscript, the sample was bilayer and all experiments were performed in room temperature). In the original work, we also used a circularly-polarized probe to simultaneously measure the shifts of X_1 and X_2 . Consequently, both excitons were influenced on differential transmission (DT), resulting in somewhat complex shapes of spectra (Fig. 3a and 3b in the original manuscript). This invoked many free parameters in the fitting procedure. In such a situation, the fitting procedure is generally non-intuitive rendering the estimation of the exciton shift (ΔE) to be overly sensitive to input parameters. For this reason, we have deleted information on the estimation of ΔE in the revised Supplementary Information: instead of plotting ΔE obtained from fits (Fig. R1-1a, corresponds to Fig. 3c in the original main text), we have plotted the measure absolute DT value at 1.529 eV (blue circles) and 1.619 eV (red circles), at which the responses are dominated by optical Stark shifts of X_1 and X_2 , respectively (Fig. R1-1b, corresponds to Fig. S7c in the revised Supplementary Information). We can see that the pump-polarization dependence of the re-measured data (Fig. R1-1b) is almost identical to that of the original ones (Fig. R1-1a).

Fig. R1-1. (a) Pump-polarization-dependent shifts of X_1 (blue) and X_2 (red) for bilayer ReS_2 at room temperature in the original main text (Fig. 3c in the original main text). (b) DT values at 1.529 eV (blue circles) and at 1.619 eV (red circles) (Fig. S7c in the revised Supplementary Information). Solid lines are fits.

We of course have provided information on the estimated ΔE in the revised main text. There, a more accurate and intuitive estimation of ΔE have been possible because the re-measured DT spectra have simple absorption-derivative-like lineshapes (Fig. R1-2a and R1-2b) requiring only one free fitting parameter for each case (details of the revised fitting procedure is described in the revised Supplementary Information S1). Similar to the original results, the pump-polarization-dependent ΔE of X_1 and X_2 follow their own spatial orientations, as shown in Fig. R1-2c and R1-2d, where ΔE values of both excitons do not vanish even when the pump polarization is orthogonal to their orientations. These behaviors are similar to the results in the original manuscript (Fig. R1-1a). In the original work, however, such behavior did not agree with the light-polarization-dependent equilibrium absorption because the excitonic oscillator strength was negligible when the light polarization was orthogonal to the direction of each exciton (Fig. R1-3a). For this reason, we mentioned the possibility of coherent coupling between optical Stark shifts of X_1 and X_2 in the original manuscript, and we now attribute this to the shared valence or conduction bands (Table. R1-1), according to Reviewer 1's suggestion. However, in contrast to the original work, the light-polarization-dependent excitonic absorption does not vanish even when the light-polarization is orthogonal to the axis of each exciton, as shown in the polar plot of Fig. R1-3b. This result is not at odds with the pump-polarization-dependent ΔE in Fig. R1-2c and R1-2d. For this reason, we have not stated the possibility of the shared valence of conduction band in the revised main text. Instead, we have presented this statement in the revised Supplementary Information, as mentioned above. We thank again Reviewer 1 for providing her/his insightful interpretation.

Unfortunately, regarding the light-polarization-dependent spectral weight of excitons (polar plots in Fig. R1-3), the discrepancy between the original and the re-measured data is unknown at this stage. This may be related to possible interlayer coupling or spatial confinement considering the different sample thicknesses in those measurements.

Alternatively, in the room temperature experiment of the original manuscript (Fig. R1-3a), the exciton spectrum having small spectral weight can be masked by background due to its broad linewidth, which can possibly lead to the vanishing oscillator strength at orthogonal light polarizations.

Fig. R1-2. (a,b) DT spectra at $\tau = 0$ fs of few-layer ReS₂ at 78 K with co-linear pump-probe polarizations. The linear probe polarization angles are $\theta = 20^\circ$ (b) and $\theta = 90^\circ$ (c). (d,e) Pump-polarization-dependent optical Stark shifts of X₁ (blue dots in (d)) and X₂ (red dots in (e)) with fixed probe polarizations of $\theta = 20^\circ$ and $\theta = 90^\circ$, respectively. The solid lines are fits. These figures have been inserted into the Fig. 3 of the revised main text.

Fig. R1-3. Polarization-dependent absorption spectra and corresponding spectral weights of Lorentzian excitonic contributions (blue dots for X₁; red dots for X₂) of bilayer ReS₂ at room temperature (a; Fig. 1b-1c in the original manuscript) of few-layer ReS₂ at 78 K (b; Fig. 1d-1e in the revised manuscript).

General remarks of Reviewer 2:

The authors report the observation of a polarization-dependent optical stark effect in ReS₂. This arises due to the anisotropic formation of excitons in this material. The observations were made using ultrafast optical pump-probe techniques, which is the same technique used in previous observations of the optical stark effect. The observation of a polarization dependent optical stark effect is new, the data and analysis is convincing, and the paper is well written. I think it is highly suitable for publication in Nature Communications.

Response:

We appreciate the time Reviewer 2 took to read our manuscript and to provide his/her thoughtful opinions. We are pleased that Reviewer 2 found that our work is highly suitable for publication in *Nature Communications*. His/her comments on the relation of the anisotropy with valley degrees have helped improve our manuscript. We have faithfully considered the comment and revised our manuscript correspondingly. Below we present our response to the Reviewer 2's comment.

Comments 2-1:

I would only suggest adding some words on how this anisotropic optical stark effect affects things in the two valleys at K and K'. This would be useful information since the previous publications on this subject have focused on the valley selectivity of the effect in this material class.

Response 2-1:

We thank the reviewer for her/his important comments. As Reviewer 2 pointed out, it is important to compare our results with previous studies on the valley selective optical Stark effect in group VI TMD materials (ref. 12, Sie *et al.*, Nat. Nanotechnol. 14, 290 (2014); ref. 13, Kim *et al.*, Science 346, 1205 (2014)). This is because all these studies, including our work, deal with selective optical Stark effect in 2D TMD materials. Although we already discussed differences between our work and previous studies in the original manuscript (line 52-58), we agree with Reviewer 2 that additional comments can be very useful for readers.

As discussed in the original manuscript, while ReS₂ is a group-VII TMD with in-plane anisotropy, materials studied previously are group-VI TMDs (WS₂ and WSe₂) which have high in-plane symmetry with a hexagonal structure. Here, the key factor of the valley selectivity is the nontrivial Berry phase at K and K' points, which is characteristic of monolayer group-VI TMDs. However, unlike group VI TMDs, the optical selectivity of ReS₂ in our work stems from linearly anisotropic excitons near Γ point due to in-plane anisotropy of crystal structure. Thus, it seems somewhat difficult to expect significant connections of the

linear anisotropy of ReS₂ with the valley degree of freedoms at K (K') points. Unfortunately, we cannot state the effect of the anisotropic optical Stark effect on valley selectivity for now. We expect that further theoretical study will find valley characteristics and Berry phases at important momentum points in ReS₂. For this reason, regarding the issue mentioned in the Comment 2-1, the only thing that we can do at this stage is to provide more specific information on the momentum positions of group-VI and group-VII TMDs. We thus have added some phrases to the revised manuscript, as shown in Table. R2-1. We hope reviewer 2 understands that we cannot offer more theoretical background, and thank again for providing her/his thoughtful suggestion.

Original	Revised
Since the valley excitons in these studies are energetically indistinguishable (line 56-57, page 3)	Since the valley excitons at K (K') point in these studies are energetically indistinguishable (line 58-59, page 3)
The absorption peaks, labeled as X ₁ and X ₂ , arise from the two lowest, energetically nondegenerate direct exciton states (line 66-68, page 3)	The absorption peaks, labeled as X ₁ and X ₂ , arise from the two lowest, energetically nondegenerate direct exciton states near Γ point²⁷ (line 70-71, page 3)

Table. R2-1. Revised sentence.

In addition to the issue raised by Reviewer 2, we have made important changes in the manuscript. We have re-measured all data sets using a few-layer ReS₂ sample at 78 K in order to improve the data quality. In addition, while we have used circularly-polarized probe in our original work, in the revised manuscript, we used linearly polarized light to detect more direct signals for the selective optical Stark effect. As a result, we have obtained more intuitive data sets, but, the main conclusion (i.e., selective anisotropic optical Stark effect) has not been changed. Please refer to the revised main text and Supplementary Information.

General remarks of Reviewer 3:

The authors performed an optical Stark effect experiment on bilayer ReS₂. They claim that two different states can exhibit relatively different energy shifts depending on the laser polarization. They show additional fluence and time dependence measurements to support the above conclusion.

I have read carefully the main text and the supplementary, and below is my review concerning (i) the novelty, (ii) the quality/clarity, and (iii) the impact of this work, which are the criteria to maintain the high-standard journal of Nature Communications.

Response:

We appreciate the time Reviewer 3 took to read our manuscript. Her/his thoughtful comments on the novelty, the data quality and the impact of our work have helped us to significantly improve our manuscript with high completeness. In order to obtain high quality data, we have performed all measurements under changed experimental conditions. We also have rewritten a considerable portion of the manuscript. Below we present our point-by-point response to Reviewer 3's comments.

Comments 3-1:

Regarding the novelty of this work, it is natural to make a comparison with the existing related works on the optical Stark effect found in the literature: Note that the optical Stark effect is already known for many years in atoms and in solids (References 1-11). Selectively tunable optical Stark effect has also been shown in transition-metal dichalcogenides (TMDs, References 12-13) and in lead-halide perovskites (Science Advances, DOI: 10.1126/sciadv.1600477), where two different exciton states can exhibit different energy shifts depending on the laser polarization. This is basically the same phenomenon that is claimed by the present author, and simply using linear instead of circular light polarization into a slightly different material is not sufficient to claim this as a new finding. In addition to bilayer ReS₂, there are many other materials that exhibit anisotropic electronic and optical properties such as carbon nanotubes and black phosphorus, from which anisotropic optical Stark effect is expected. In this perspective, the present manuscript is too similar with existing works, and it lacks the novelty required for Nature Communications.

Response 3-1:

We thank Reviewer 3 for providing her/his thoughtful comments on the novelty of our work. In the manuscript, we demonstrated that the excitonic Stark effect in ReS₂ can be selectively tuned by manipulating the polarization angle of linearly polarized light. This was possible

because the lowest two excitonic transitions in ReS₂ have different linear anisotropy, which stems from reduced symmetry of the crystal structure.

In comment 3-1, Reviewer 3 claimed that our work lacks novelty partly because recent papers have already reported the light-polarization selectivity of the excitonic optical Stark effects in other materials, such as monolayer group-VI six TMD and 2D perovskites (Sie *et al.*, Nat. Nanotechnol. 14, 290 (2014); Kim *et al.*, Science 346, 1205 (2014); Giovanni *et al.*, Sci. Adv. 2, e1600477 (2016)). Reviewer 3 also pointed out that that linearly polarization of the excitation light in our work is the only difference compared to these previous studies where circularly polarized light were used. However, we strongly disagree with Reviewer 3's opinion. The main novelty of our finding does not simply lie in the use of the different type of light polarization. Because the novelty is one of the most important standards in Nature communications, below we address in great detail by breaking down the issue in several different viewpoints.

Fig. R3-1. Representative examples of light-polarization-selective optical Stark effect in early papers. (top, Choi *et al.*, Phys. Rev. B 65, 155206 (2002)) Schematic description of polarization-dependent optical Stark effect and corresponding differential absorption spectra for co-polarized ($\sigma+$, $\sigma+$) and counter-polarized ($\sigma+$, $\sigma-$) pump-probe configurations in GaN. (middle, Joffre *et al.*, Phys. Rev. Lett. 62, 74 (1989)) Schematic for the optical Stark effect in GaAs multiple quantum well. Right upper panel shows equilibrium absorption. Right lower panel shows differential absorption due to optical Stark shifts for co-polarized (solid line) and counter-polarized (dotted line) pump-probe configurations. (bottom, Gupta *et al.*, Science 292, 2458 (2001)) Schematic description of polarization-dependent optical Stark effect in ZnCdSe quantum well. Right panel displays light-intensity-dependent Stark shift (ΔE) for co-polarized (blue) and counter-polarized pump-probe configurations (red).

First of all, it should be noted that, prior to those papers that Reviewer 3 mentioned, there have already been many published studies on the polarization selectivity of the excitonic optical Stark effects (e.g., Joffre *et al.*, Phys. Rev. Lett. 62, 74 (1989); Gupta *et al.*, Science 292, 2458 (2001); Choi *et al.*, Phys. Rev. B 65, 155206 (2002)). As summarized in Fig. R3-1, these early studies showed that the optical Stark shift can efficiently occur only when the polarizations of pump and probe beams share the same circular polarization, whereas the effect is significantly weakened in the counter-polarized configuration. Such polarization-selectivity stems from the light-polarization-dependent optical selection rules. Considering only the phenomenological results, main results in the early studies are almost the same as the recent papers that Reviewer 3 mentioned. Similar to the early works, excitonic optical Stark shift in monolayer group VI TMDs and 2D provskites also take place only when the pump and probe are co-polarized due to the light-polarization-dependent optical selection rules. Of course, there is a small phenomenological difference; unlike early papers, the Stark shift in the recent works can be almost completely suppressed in counter-polarized pump-probe configuration, owing to their non-shared electronic or hole states in excitonic transitions. Such a difference is simply related to the efficiency of the polarization selectivity, but, it cannot be a critical liability for the novelty of the recent works.

If we follow Reviewer 3's opinion, it seems that the recently reported optical Stark effect in group VI 2D TMD should also significantly lose their novelty since the basic physics is exactly the same as the early works. However, there is no doubt that these papers have a great novelty, confirmed by the fact that both of them were published in top journals (Nature Nanotechnology and Science). As far as we understand, the key novelty of these works lies not in the polarization selectivity of the Stark effect, but in the selective optical control of "valley" states which are newly emerged quantum degrees of freedom in the monolayer group VI TMDs. In other words, the phenomenological similarity cannot reduce the novelty of the associated studies. Viewed in this way, our work definitely has novelty. As described in the original manuscript, the two lowest excitons in ReS₂ are linearly polarized due to its reduced in-plane structural symmetry, and, more importantly, their optical selection rules manifest completely different light-polarization dependence. The fundamental physics of such unique excitonic characteristics are obviously different from other materials in previous works, such as valley-selective optical transition in monolayer group VI TMDs (determined by the nontrivial Berry phase at K (K') valleys and angular momentum of atomic orbitals) and spin-selective optical selection rules in conventional semiconductors of early studies. Moreover, there have been no reported studies on the optical Stark effect of linearly anisotropic excitons with distinct optical selection rules.

Of course, ReS₂ is not the only material exhibiting anisotropic property of excitons; there are several materials that anisotropic excitonic optical Stark effect can be expected (such as carbon nanotubes (CNTs) and black phosphorus (BP)), as Reviewer 3 mentioned. However, such studies have not been reported yet. Although there is one paper reporting the optical Stark effect in CNTs (Song *et al.* Appl. Phys. A 96, 283 (2009)), no polarization dependence was measured. More importantly, unlike ReS₂, both of these materials lack polarization-dependent exciton selectivity so that light-polarization-controlled exciton-selective optical Stark effect cannot be expected. For CNTs, since anisotropic optical transition is simply determined by their one-dimensional structure, all excitonic transition should have the same polarization dependence. For BP, there is only one prominent excitonic transition with distinct anisotropy (Wang *et al.* Nat. Nanotechnol. 10, 517 (2015)). After thorough literature searches, no single literature exists reporting the optical Stark effect in BPs. If one says that the experimental result is not novel simply because it is “expected”, then the argument went too far from reality, and there are virtually no novel scientific experiments other than the theoretical expectation. Thus, among other anisotropic materials that Reviewer 3 mentioned, our observation is (arguably) very original, and ReS₂ itself is an interesting material platform for testing the polarization-dependent exciton selective control of optical Stark effect. Regarding this issue, we have added following sentences in the revised main text.

- Added sentences in the revised manuscript (Discussion section)

Of course, group VII TMDs are not the only material family exhibiting anisotropic property of excitons; there are several systems possessing anisotropic excitonic properties (such as carbon nanotubes (CNTs) and black phosphorus (BP))^{12,30-32}. However, both of them lack polarization-dependent exciton selectivity so that energy-selective optical Stark effect cannot be expected. For CNTs, since the anisotropy of excitonic transition arises simply from the geometrical alignment, all excitonic transitions should have same polarization dependence¹². For BP, there is only one prominent excitonic transition with distinct anisotropy³⁰. Thus, group VII TMDs are ideal material platforms for testing the energy selective control of the excitonic optical Stark effect.

Additionally, one paper mentioned by Reviewer 3 (Giovanni *et al.*, Sci. Adv. 2, e1600477 (2016)) was published just after we have submitted our manuscript to Nature Communications. We even did not notice whether this paper was already submitted or not. Thus, it is inappropriate to compare our work with the paper mentioned when discussing novelty; this paper was published just before we received the Reviewers’ reports, giving us no chance in telling how novel the work is compared to our work. However, even if the comparison is acceptable, we want to emphasize that our work still has novelty because the

background physics is completely different, as discussed above. We have added this paper to the reference list in the revised main text.

Comments 3-2:

More fundamentally, note that the two states investigated in this study are already different in energy by 50 meV. Hence, any attempt to further shift their relative energies by merely 1 meV has only little significance for fundamental science and applications. This situation is different from the selective energy shift in TMDs and perovskites above because the two states are originally identical in energy and protected by a certain symmetry. Hence, shifting their relative energies is of significant interest in fundamental science and applications, which is not the case for bilayer ReS₂.

Response 3-2:

We fully agree with Reviewer 3 that lifting of degenerate states using selective energy shift has fundamental significance (Sie *et al.*, Nat. Nanotechnol. 14, 290 (2014); Kim *et al.*, Science 346, 1205 (2014)). However, controlling energy shifts of non-degenerate state also has importance in terms of energy- or frequency-selective modulations. Many literatures have mentioned that the applicability of the excitonic optical Stark effect lies mainly in ultrafast optical modulating, switching and information processing devices. Thus, there is no doubt that relevant optoelectronic devices can have higher functionality and degree-of-freedom if energy-selective control is possible. So far, however, most relevant studies have focused only on the optical Stark shift of the lowest exciton state (e.g., heavy-hole exciton in GaAs-based quantum wells, A-exciton in group VI TMDs). Although several studies dealt with optical Stark shifts of higher states, no energy-selective control has been reported. This is because there have been no methods in selectively tuning the Stark shift of higher exciton states. In Comment 3-6, Reviewer 3 mentioned that the optical Stark shift of a higher exciton state can be done simply by tuning pump photon energy. However, such a modulation method is not proper in common situations because if we raise the pump photon energy close to the higher-lying exciton state over the lower-lying state, it will cause generation of significant magnitude of real excitons and free carriers, which obstruct the observation of the optical Stark effect. For this reason, the maximum photon energy of the pump beam has always been lower than that of the lowest exciton resonance energy in every case, and tuning the pump photon energy has not been utilized in controlling the energy-level-selective optical Stark effect of excitons.

In the revised process, we have obtained very intuitive signals for completely selective shifts of excitons. Although the magnitudes of shifts are about 1 meV, the data clearly shows frequency-selective applicability in modulations or switches. Please refer to response 3-3.

Regarding this issue, we have modified the schematic description of the optical Stark effect in ReS_2 (Fig. 1d-1f in the original main text), in order to more clearly reveal the significance of our work. In the revised manuscript, we compare the optical Stark effect in conventional semiconductors and ReS_2 , as shown in the right panel of Fig. R3-2. Note that in the revised version (Fig. R3-2b), we deleted the unselected exciton levels for the selective shifts of X_1 (middle panel) and X_2 (right panel). This is because, unlike the original work (where circularly polarized probe had used to simultaneously measure the shifts of both excitons), we used a linearly polarized probe beam to selectively measure the blue-shift of each exciton state in the revised work. Please refer to the Response 3-3 for details of the changed experimental conditions.

Fig. R3-2. Revised schematics in Figure 1. (a) Original schematic illustrating the selective optical Stark effect in ReS_2 (Fig. 1d-1f in the original manuscript). (b) Revised figures. Comparison between conventional semiconductors and ReS_2 is added (Fig. 1a-1b in the revised manuscript).

Comments 3-3:

Regarding the quality of this work, I think the measured DT spectra (Fig 2c, Fig 2e, Fig 3a) can still benefit from (i) improving the signal-to-noise ratio and from (ii) acquiring finer interval data points, as compared to the better data quality in the earlier works mentioned above. Besides, these DT spectra do not show a straightforward interpretation of an energy shift, where a simple derivative-like curve should be expected. This is because the two states are not well separated in energy, with energy separation that is comparable to their peak widths. I have no doubt that the optical contribution from the optical Stark effect does exist, as the authors have provided their best efforts to show it in their analysis. But again the compromising data quality makes it difficult to disentangle the contributions from possible coherent spectral oscillations (Phys. Rev. Lett. 59, 2588 (1987), Optics Letters 13, 276 (1988)) or from other long-lived dissipative processes. This also makes difficult to accurately determine the magnitude of the energy shift that, as of now, can be too sensitive to the input fitting parameters. I am afraid that the lack of clarity in the data may be confusing for some readers from interdisciplinary background.

Response 3-3:

We thank the reviewer for his/her very important comments. We agree that better data quality can improve the accuracy of interpretation of the measured DT spectra. For this purpose, we have re-measured full data sets under changed experimental conditions. As a result, we have obtained high quality data and simplified the fitting procedure. Details of improvements are described below.

i) We changed the sample from the bilayer to the few-layer (7-8 layers) ReS_2 , as shown in Fig. R3-3. This is because thicker samples generally produce larger pump-probe signals. In addition, owing to increased surface-to-volume ratio in the thick sample, surface defects can be reduced so that unwanted noise can also be lowered and coherent interaction like the optical Stark effect takes place more efficiently. The signal-to-noise ratio (SNR) was indeed significantly enhanced in the revised version, as shown in the discussion below.

Fig. R3-3. Left: optical image of the bilayer ReS_2 (inset of Fig 1a in the original manuscript). Scale bar 25 μm . Right: optical image of few-layer ReS_2 (inset of Fig. 1c in the revised manuscript). Red graph shows the AFM height profile along the red dashed line. Scale bar, 15 nm.

ii) In the revised version, all measurements were performed in low temperature (78 K). In the original manuscript, experiments have been performed in room temperature so that absorption linewidths of two excitons are somewhat broad compared to the spectral distance between their resonance energies. On the contrary, the linewidths of absorption resonances at 78 K are significantly decreased, as shown in Fig. R3-4. Such temperature-dependent narrowing of linewidths well agrees with a previously reported paper [Ho *et al.* Phys. Rev. B 55, 15608 (1997)]. Now we can say that X_1 and X_2 are well separated at 78 K in the spectral domain, as displayed in Fig. R3-5; the spectral distance between X_1 and X_2 excitons (0.033 eV, black arrow) are about 1.65 times larger than the sum of their half linewidths (0.011 eV + 0.009 eV, green arrows). Since separation of excitons is an important issue in terms of energy-selective optical Stark effect, we have inserted Fig. R3-5 and related sentences into

the revised Supplementary Information S4. Please refer to the revised Supplementary Information.

Fig. R3-4. Light-polarization-dependent equilibrium absorption spectra of the bilayer ReS₂ at room temperature (Fig. 1b in the original manuscript) (left) and few-layer ReS₂ at 78 K (Fig. 1d in the revised manuscript) (right).

Fig. R3-5. Spectral distance between X₁ and X₂. Blue and red lines are back-ground absorption spectra at 78 K with light polarization of $\theta = 20^\circ$ (blue) and $\theta = 90^\circ$ (red). Black arrow shows spectra distance between X₁ and X₂. Green arrows indicate their half linewidths. This figure is inserted into Supplementary Information Section 4.

More importantly, the narrowed absorption linewidths made the data quality of DT much clearer, because the amplitude of DT signal originating from the optical Stark effect is proportional to the derivative of absorption resonance spectrum. For example, we can see that the DT time-traces at 78 K (right panel in Fig. R3-6, Fig. 2d in the revised manuscript) show higher SNR than those measured at room temperature (left panel in Fig. R3-6, Fig. 2d in the original manuscript).

Fig. R3-6. DT time-traces of the bilayer ReS₂ at room temperature (Fig. 2d in the original manuscript) (left) and few-layer ReS₂ at 78 K (Fig. 2d in the revised manuscript) (right). Corresponding probe photon energies are indicated.

iii) We used linearly polarized probe beams to selectively measure the DT response for a certain exciton state in the revised manuscript. In the original manuscript, we have used a circularly polarized probe in order to simultaneously detect the optical Stark shifts of both excitons and their pump-polarization dependence. Although such experimental scheme provided valuable results, the forms of spectra (Fig. R3-7a) are somewhat non-intuitive because their oscillatory shapes do not follow the simple absorption-derivative-like shape. We also agree that the simultaneous measurement of shifts in both exciton states makes it difficult to distinguish the response of one exciton to the other. We guess that this issue is also closely related to the coherent spectral oscillations mentioned in Comment 3-3.

Fig. R3-7. Exciton-selective optical Stark effect controlled by light-polarization. **(a)** DT spectra of linearly polarized optical excitations at room temperature with two important polarization angles of $\theta = 20^\circ$ (blue) and 80° (red), at which the pump light is closely aligned to X₁ and X₂, respectively (Fig. 3a in the original manuscript). Here, probe is circularly polarized. Sample is bilayer ReS₂. **(b,c)** Equilibrium absorption spectra of few-layer ReS₂ at 78 K with light polarization angle of $\theta = 20^\circ$ (top panel in **(a)**) and $\theta = 90^\circ$ (b) (top panel in **(b)**). Black dots (middle panels) and gray dots (bottom panels) are corresponding DT spectra at $\tau = 0$ fs with co-polarized and cross-polarized pump-probe configurations, respectively. Probe polarization angles are fixed at $\theta = 20^\circ$ (**b**) and $\theta = 90^\circ$ (**c**). Pump photon energy is 1.44 eV with fluence of $230 \mu\text{J cm}^{-2}$. In **(a-c)**, the blue- (red-) shaded area represents the spectral region where DT signal is dominated by the shift of the X₁ (X₂) state. **(d)** Schematic illustration of DT spectrum (black line) due to Stark shift for an exciton state. Gray and blue lines indicate unperturbed and blue-shifted absorption resonances, respectively.

To resolve this issue, unlike the original work, we have selectively measured the optical Stark shift of a certain exciton by using a linearly polarized probe. First, in order to measure the Stark shift of X_1 , the probe polarization angle has been fixed at $\theta = 20^\circ$, at which X_1 dominates the optical response and X_2 has negligible oscillator strength (see the equilibrium absorption spectrum in the top panel of Fig. R3-7b). Under this condition, we have observed that DT response of a co-linearly polarized pump shows simple absorption-derivative-like shape (middle panel, Fig. 3a) only at the spectral region dominated by X_1 (blue-shaded area), which indicates selective optical Stark effect of X_1 . This result is quite intuitive, and very easy to estimate the magnitude of the shift in the exciton level. The amplitude of Stark signal becomes small when the pump is orthogonally polarized to the probe (bottom panel of Fig. R3-7b), which indicates reduced blue-shift which agrees well with the original work. In a similar manner, we have selectively measured the optical Stark shift of X_2 at a fixed probe polarization of $\theta = 90^\circ$ (top panel in Fig. R3-7c). We can clearly see the selective optical Stark shift of X_2 , as shown in the middle panel in Fig. R3-7c. Similar to the X_1 's response, it shows decrease in amplitude in the cross-polarized pump-probe configuration (bottom panel in Fig. R3-7c). Compare to the original work, the re-measured data has higher SNR and spectral resolution.

We have inserted these results into Fig. 3a-3b and the section “Energy-selective optical Stark effect” in the revised main text. We also added a schematic description of derivative-like absorption shape of DT spectrum (Fig. R3-7d) to the revised manuscript (Fig. 2b), in order to help readers from interdisciplinary background understand the measured spectra. Please refer to the revised manuscript.

iv) Although the re-measured DT spectra (Fig. R3-6b and R3-6c) has a simple and intuitive absorption-derivative-like shape, direct estimation of the energy shift (ΔE) is still somewhat inappropriate. This is because the DT response at $\tau = 0$ fs is affected by the pump-excited real carriers, generated by two-photon- or phonon-mediated-absorption of pump photons (Knox *et al.* Phys. Rev. Lett. 62, 1189 (1989)). Such a mixed response is corroborated in the DT trace in the right panel of Fig. R3-6, where a spike-like peak near $\tau = 0$ fs due to the optical Stark effect is followed by long-lasting signals originating from pump-generated carriers. This issue has been a common problem in many relevant studies on the excitonic optical Stark effect in solids (e.g., Von Lehmen *et al.* Opt. Lett. 11, 609 (1986); Knox *et al.* Phys. Rev. Lett. 62, 1189 (1989); Sie *et al.* Nature Nanotechnol. 14, 290 (2015)). To resolve this problem, we have fit the transient DT spectra at $\tau = 400$ fs (at which no Stark shift is seen) with corresponding fitting parameters to estimate ΔE in the original work (Supplementary Information S2). However, such methodology is overly sensitive of the input fitting

parameters, as Reviewer 3 pointed out. For this reason, we have simplified the fitting procedure in the revised manuscript. An example is shown below.

Figure R3-8 shows the estimation procedure of shift of X_1 with co-linear pump-probe polarization configuration in the revised manuscript. As discussed above, unlike the original work, we selectively measured the shift of X_1 exciton. Thus, the magnitude of the shift can be estimated simply by comparing the measured DT lineshape with the $(\partial A / \partial E)/(1 - A)$ spectrum scaled by ΔE (where A is the equilibrium absorption). Now let us analyze the measured DT spectra. While the DT spectrum at $\tau = 0$ fs shows typical absorption-derivative-like spectrum (Fig. R3-8b), that measured directly after the optical Stark shift (at $\tau = 350$ fs, Fig. R3-8c) shows typical bleaching-like shape of X_1 transition. These results well agree with the DT time-trace in Fig. R3-8a. In order to distinguish the bleaching effect from the Stark shift, we subtract the DT at $\tau = 350$ fs (Fig. R3-8c) from the DT at $\tau = 0$ fs (Fig. R3-8b), producing an almost symmetric shape, as indicated by black dots in Fig. R3-8d (similar approach is shown in the paper ‘Sie *et al.* Nature Nanotechnol. 14, 290 (2015)’). This background-subtracted DT spectrum now closely follows the lineshape of $(\partial A / \partial E)/(1 - A)$ (red line in Fig. R3-8d), which enables us to estimate ΔE by comparing two graphs in Fig. R3-8d. In this way, we could obtain the shift of the energy level by using only one free parameter (ΔE). We inserted Fig. R3-8 and related discussion into the revised Supplementary Information S1.

Fig. R3-8. (a) DT trace at the fixed probe photon energy of 1.526 eV (at 78 K). Polarizations of pump and probe are both $\theta = 20^\circ$. (b,c) Corresponding DT spectrum at $\tau = 0$ fs (b) and 350 fs (c). (d) Black dots are background-subtracted DT spectrum (left axis) and red solid line is $(\partial A / \partial E)/(1 - A)$ scaled by ΔE (right axis).

Finally, we transferred all results in the original main text to the Supplementary Information in the revised version because they agree well with the re-measured ones in the revised main text. However, in order to reflect Reviewer 3’s concern, we deleted the estimation of ΔE . As shown in the right panel of Fig. R3-9, we plotted the pump-polarization-dependent absolute

DT values of two different probe photon energies of 1.529 eV and 1.619 eV, at which the DT response is dominated by X_1 and X_2 , respectively. We can see that the signal at 1.529 eV (1.619 eV) directly follows the polarization-dependent optical selection rule of X_1 (X_2) exciton. Despite the absence of the estimation of ΔE , these results can be a good Supplementary Information for the main results in the revised manuscript, considering that the DT signals arise from the optical Stark effect. Please refer to the revised Supplementary Information S5 for details.

Fig. R3-9. (left) Pump-polarization-dependent DT spectra of bilayer ReS2 at room temperature. This graph is same with Fig. 3b in the original main text. (right) Corresponding absolute DT values at 1.529 eV (blue circles) and at 1.619 eV (red circles). Solid lines are fits.

Comments 3-4:

The authors emphasize on the word "selectivity" in this work, but according to Figure 3c the selectivity is only up to a factor of 1.75-to-0.50. This is a rather poor contrast as compared to other existing works, and it is rather impractical for applications, in contrast to the authors' claim.

Response 3-4:

We thank Reviewer 3 for her/his very important comment on the selectivity. In the original work, while we have used polarization-controlled linear pump pulses to excite the sample, the probe had been a circularly polarized pulse (having the whole directions of electric fields). The reason for this was to experimentally exclude the contribution of probe on the polarization dependence, and to focus on the pump polarization dependence of the anisotropic optical Stark effect of excitons as described in the original manuscript (line 94-99, page 4). As a consequence, resultant DT spectra should have contributions from both excitons, as shown in the Fig. R3-7a and the left panel of Fig. R3-11 (Fig. 3a and Fig. 3b in the main text, respectively). In such a situation, the only way to distinguish the contribution of each exciton is to compare of the energy shift of these states. Figure R3-12a shows the

corresponding graph in the original main text (Fig. 3c in the original manuscript); the selectivity looks only up to a factor of 1.75-to-0.50, as Reviewer 3 pointed out.

In the revised manuscript, we have measured the optical Stark shift of each exciton state in a completely selective manner. This was possible since we have used a linearly polarized probe to pre-exclude the response of the unselected exciton, as explained in detail in the Response 3-3. We repeatedly plotted the corresponding results, as shown in Fig. R3-10b (same with the middle panels of the Figs. R3-7b and R3-7c), in order to clearly reveal the improvement. There, obviously we can see that only the selected exciton shows optical Stark effect-induced DT response: when X_1 (X_2) is selected, absorption-derivative-like response is observed only at the spectral region of X_1 (X_2), as indicated by blue- (red-) shaded area in Fig. R3-10b (Fig. R3-10c). Thus, these results themselves clearly show the “selectivity” without the necessity to compare magnitudes of their Stark shifts. This interpretation is further supported by the fact that two excitons are well separated in the spectral domain (Fig. R3-5). In this way, energy-selective measurement of excitonic optical Stark effect is possible. Regarding this issue, we added relevant sentences to the revised manuscript as follows.

- Added sentences in the revised manuscript (line 149-158, page 7)

These results enlighten us of significant benefits of ReS_2 in terms of selective optical control of excitons. Firstly, as shown in the middle panels of Fig. 3a and 3b, it is possible to measure the shift of a certain exciton state in a completely exclusive manner, indicating high *exciton-selectivity*. More importantly, the results also reveal *energy-selectivity*, considering that the two exciton states possess well-separated energy levels (note that the spectral distance between the two exciton resonances are larger than the sum of their half linewidths, see Supplementary Information S4). In particular, the higher-lying exciton state (X_2) can be selectively modulated without being disturbed by the lower-lying exciton (X_1) (see Fig. 3b). Such unique functionality is absent in other materials, such as semiconductor quantum wells, carbon nanotubes and group VI TMDs. Schematics in Fig. 1b summarize these findings.

In addition, we of course present the pump-polarization-dependent shifts of excitons in the revised manuscript, as shown in Fig. R3-10d and R3-10e. Note that, unlike the original version (Fig. R3-10a), we plot the shifts of X_1 (Fig. R3-10d) and X_2 (Fig. R3-10e) in separated figures. This is because the original work had simultaneously measured the shifts of both excitons in the same condition, while different probe polarizations were used in the revised measurements ($\theta = 20^\circ$ and 90° in Fig. R3-10d and R3-10e, respectively). There, we

can see that the pump-polarization-dependent shifts of each exciton directly follow their own spatial orientations, agreeing well the original results. We have inserted Fig. R3-10d and R3-10e into the revised manuscript (Fig. 3c and 3d).

Fig. R3-10. (a) Pump-polarization-dependent shifts of X_1 (blue) and X_2 (red) for bilayer ReS_2 at room temperature in the original main text (Fig. 3c). (b,c) DT spectra at $\tau = 0$ fs of few-layer ReS_2 at 78 K with co-linear pump probe polarizations. The probe polarization angles are $\theta = 20^\circ$ (b) and $\theta = 90^\circ$ (c). (d,e) Pump-polarization-dependent optical Stark shifts of X_1 (blue dots in (d)) and X_2 (red dots in (e)) with fixed probe polarizations of $\theta = 20^\circ$ and $\theta = 90^\circ$, respectively. The solid lines are fits.

Comments 3-5:

Regarding the impact of this work, "Science-wise" the anisotropic property of this material is already known and the polarization-selective optical Stark effect is already demonstrated in previous works, "Applications-wise" it is a little difficult to say because the observed effect (1 meV) is much smaller than the linewidth and the thermal energy at room temperature. I think currently it is rather important to study the equilibrium phenomena of this material more rigorously. For example, the equilibrium electronic structure of ReS_2 still suffers from controversial reports on whether the lowest energy gap corresponds to a direct or an indirect transition (References 18 and 23), or whether the interlayer coupling is really insignificant (References 18 and Nano Lett. 16, 1404 (2016) etc.). This controversy could affect the data interpretation of the present work. This is extremely important especially considering the high standard Nature Communications that maintains the novelty, the high data quality, and the correct interpretation.

Therefore, I cannot recommend the publication of this work in Nature Communications. But I think the authors can still consider submitting their works in a more specialized journals, possibly in the ACS or AIP journals.

Response 3-5:

First, “Science-wise”, we have already provided sufficient ground for the novelty of our work. Please refer to the Response 3-1.

Second, “Application-wise”, we agree with Reviewer 3 that the magnitude of the exciton shift (ΔE) is somewhat small compared to the broad linewidth in the original manuscript. However, in the revised version, the linewidths are significantly reduced at low temperature (right panel of Fig. R3-4), enhancing the efficiency of the Stark effect. In order to evaluate the efficiency of ReS₂ precisely, we have compared the strength of this effect in ReS₂ with that measured in other TMD materials. The strength of the optical Stark effect is defined by $S = (\Delta E \times \delta) / 2\varepsilon^2$ (where δ is pump detuning and ε is the intensity of pump field). Since the parameter S is generally regarded as a material’s intrinsic property, many paper have used it to compare the efficiency of the optical Stark effect, instead of ΔE (Kim *et al.* Science 346, 1205 (2014); Song *et al.* Appl. Phys. A 96, 283 (2009)). In our case, S are $\sim 17 \text{ D}^2$ for X₁ and $\sim 15 \text{ D}^2$ for X₂ at co-linear pump-probe polarization configurations (where ΔE for X₁ (X₂) is 1.4 meV (0.8 meV) as shown in Fig. R3-10d (Fig. R3-10e), δ for X₁ (X₂) is ~ 90 meV (~ 140 meV), and ε is ~ 93 MV/m). These S values are of the same order of magnitude as that of group VI TMD ($\sim 45 \text{ D}^2$, Kim *et al.* Science 346, 1205 (2014)) at similar experimental temperatures. We have added related sentences to the revised manuscript as follows.

- Added sentences (Line 180-183, page 8)

The strengths of the optical Stark effect ($S = \Delta E_i \times (\hbar\omega_x - \hbar\omega_{\text{pump}}) / 2\varepsilon^2$)^{12,14} for X₁ and X₂ are about $\sim 17 \text{ D}^2$ and $\sim 15 \text{ D}^2$ at co-linear pump-probe polarization configurations, respectively. These values are of the same order of magnitude as that of group VI TMD ($\sim 45 \text{ D}^2$; ref. 14).

In addition to above issues, Reviewer 3 pointed out that study on equilibrium phenomena of ReS₂ should be performed more rigorously before discussing the optical Stark effect. We agree with Reviewer 3 that equilibrium characteristics of ReS₂ have not been fully understood yet. As mentioned in the Comment 3-5, whether the lowest energy gap is direct or indirect is indeed an ongoing debate (ref. 18. Tongay *et al.*, Nature Commun. 5, 3252 (2014); ref. 23. Aslan *et al.*, ACS Photonics 3, 96 (2016)): while the former study claimed that ReS₂ remains having a direct gap from bulk to monolayer, the latter paper claimed that an emission peak lower than exciton resonance in bulk was assigned to indirect gap transition. However, despite such a discrepancy, there is no doubt that the exciton resonances we measured correspond to direct transitions. Thus, it seems that the occurrence of excitonic Stark shifts

and their polarization dependence in our work are not expected to have significant connections with the possible presence of the lower-lying indirect transition. Of course, the lower-lying indirect state can behave as a relaxation channel of photoexcited real carriers in the lowest exciton state, as discussed in the previous study (Aslan *et al.*, ACS Photonics 3, 96 (2016)). However, considering that the optical Stark effect does not originate from pump-generated real carriers, it is hard to expect that this issue will be critical in our study. In addition to this issue, another important topic debated in ReS₂ is the presence of strong interlayer coupling and its influence on the layer number-dependent optical characteristics (ref. 18. Tongay *et al.*, Nature Commun. 5, 3252 (2014); He *et al.*, Nano Lett. 16, 1404 (2016)). However, although it is a very important issue, the layer number dependence of the optical Stark effect is not of interest in our manuscript because the anisotropic excitonic transitions occur regardless of the sample thickness. Indeed, the observed phenomena in our work have been successfully explained without concerning the issues Reviewer 3 mentioned. Thus, it is somewhat hard to accept that there is a significant connection between the debating issues in ReS₂ and the accuracy of our interpretation.

Comments 3-6:

Also, it would be good to use less excessive phrase in the revised manuscript:

- The phrase "multiple energy levels" is mentioned several times (Line 30, 58, 82, 150), while only two energy levels are relevant in this work. This can be potentially confusing because the readers would expect a multiple number of energy levels like 5 or more.
- Line 46, 57, 82, "So far there has been no strategy for accessing multiple energy levels of excitons in a selective manner." However, selective tuning of two different excitons has been demonstrated in TMDs and perovskites above. Also, manipulating different energy levels can also be done simply by tuning the excitation photon energy.

Response 3-6:

We agree that the phrase “multiple energy levels” may lead to misunderstanding. Thus, we have revised all sentences including that phrase, as shown in Table. R3-1.

Original	Revised
there has been no selectivity in controlling multiple energy levels (line 31, page 2)	most studies have focused on the optical Stark effect of only the lowest exciton state due to lack of energy-selectivity (line 30-31, page 2)
Our finding provides a novel methodology for selective control of multiple energy states of excitons (line 38-39, page 2)	Our finding provides a novel methodology for selective control of multiple energy states of excitons ultrafast energy-selective control of exciton states. (line 37-38, page 2)

there has been no strategy for accessing multiple energy levels of excitons in a selective manner (line 46-47, page 2)	there has been no strategy for energy-selective control of exciton states. (line 45-46, page 2)
it can be said that no experimental approaches for controlling multiple energy states of excitons have been made (line 57-58, page 3)	it can be said that no experimental approaches for energy-selective optical Stark effect of excitons have been made. (line 59-61, page 3)
The Stark shift for X_1 and X_2 follow obviously different light-polarization dependence, offering a foundation for selective control of multiple energy levels in excitonic systems (line 81-82, page 4)	We gradually tune the Stark shift for X_1 and X_2 , which obviously has different light-polarization dependence. (line 85-86, page 4)
this finding provides a foundation for selective optical control of multiple energy levels in the associated excitonic systems. (line 149-151, page 7)	deleted

Table. R3-1. Revised sentences.

Regarding the selective tuning of exciton states, we have already offered sufficient reasons that explain why the selectivity found in our work is different from those in previous studies. Please refer to the Comment 3-1. In addition, as discussed in the Comment 3-2, tuning photon energy of the pump is not a proper way to manipulate different energy levels because significant excitation of real carriers obstructs the measurement of the Stark shift and lowers the purity of this effect when the pump photon energy is higher than that of the lower-lying state. Of course, excitation with photon energy lower than the lower-lying state also cannot selectively modulate the higher-lying state in conventional materials since shift of the lower-lying exciton will be much larger than that of the higher-lying exciton. Thus, the excitonic optical Stark effect in previous studies definitely lack energy-selectivity. Recent works on 2D TMDs and perovskites also lacks the energy-selectivity because the exciton levels are energetically degenerated. We have stated this point in the revised manuscript as the original one. Please refer to second and third row in Table. R3-1.

Comments 3-7:

In particular, the following sentences in the current manuscript are stretching too far into pseudoapplications because the observed effect is too small:
- Line 37, 163, 169, "... we finally reveal a new applicability of ReS2 for modulating optical transmittance in the real-time domain." Note that the 1 meV shift is much smaller than the linewidth, and much smaller than the thermal energy at room temperature, thus rendering this effect impractical for the functionalities mentioned in the conclusions.

Response 3-7:

We agree with Reviewer 3 that the applications regarding modulation transmission in the real-time domain is somewhat far-stretched. In order to focus on the selective optical Stark

effect, we have deleted the related sentences in the original manuscript (line 36-38, page 1; line 163-170, page 7 in the original main text).

However, for the applicability of our findings, we have achieved significant improvement in the linewidth by decreasing experimental temperature (right panel of Fig. R3-4), which makes two exciton state well-separated (Fig. R3-5). It also has enabled completely selective measurements of excitons in the frequency domain via detection using a linearly polarized probe (Fig. R3-7b and Fig. R3-7c). Moreover, as discussed in Response 3-5, we have shown that the strength of the optical Stark effect in ReS₂ is of the same order of magnitude as that of group VI TMD. Base on such improvements, we believe that ReS₂ has a potential applicability for ultrafast frequency-selective optical modulators or switches.

REVIEWERS' COMMENTS:

Reviewer #1 (Remarks to the Author):

The authors have addressed to my question and I agree to publish on Nature Communication.

After reading the revised version, I have searched the literature on this material. It seems quite many important fundamental questions have not been addressed such as the origin of optical transitions in comparison to the band structure, which was pointed out by the third referee. However, that will be beyond the scope of the current work. I strongly encourage the authors to work on those for the field of research on this material.

Reviewer #3 (Remarks to the Author):

In the revised manuscript, the authors replaced some figures with new data. The data look better, i.e. narrower linewidth, more apparent derivative-like features. The authors also rewrote the manuscript in the attempt to make a more convincing case about the novelty of the work. I must say the writing is improved, and my comments are addressed quite satisfactorily. There is still one argument that I disagree. For example, in the rebuttal, the authors tried to compare the optical Stark effect (OSE) works between those in valley TMDs and GaAs QWs, and mentioned that a clear polarization-selective OSE has been done years ago in GaAs system. This is not quite the case however. In TMDs, the two valleys can be controlled separately by two circularly polarized light because they are independent due to the finite and opposite Berry curvatures. In GaAs, there will always be mixtures of states being induced with circularly polarized light because GaAs system is more like a mixture of s- and p-like orbitals in atomic systems. That is, these states are very close to each other in energy, and also not quite a clean system for such experiments, i.e. regardless of the pump polarization, there will always be a finite OSE for a given probe polarization, even theoretically.

I am still fixed to my previous standpoint regarding the novelty. However, I must admit that the revised manuscript is much more improved in terms of data quality and writing. I do foresee that this work can be of significant interests to other researchers in the field. Therefore, I think this work can be suitable for a publication in Nature Communications.

Point-by-point responses to the issues raised by the reviewers

Reviewer 1 comment:

The authors have addressed to my question and I agree to publish on Nature Communication.

After reading the revised version, I have searched the literature on this material. It seems quite many important fundamental questions have not been addressed such as the origin of optical transitions in comparison to the band structure, which was pointed out by the third referee. However, that will be beyond the scope of the current work. I strongly encourage the authors to work on those for the field of research on this material.

Response:

We appreciate the time that Reviewer 1 took to read the revised manuscript and responses to reviewers. We are quite pleased that Reviewer 1 agreed to our work being published on Nature Communications. As Reviewer 1 pointed out, various fundamental issues of ReS₂ are still under debate, such as rigorous origin of excitonic transitions and significance of interlayer coupling. We hope our work will contribute to address such issues, and we are going to continue to explore optical properties of this material. We thank again for his/her comment.

Reviewer 3 comment:

In the revised manuscript, the authors replaced some figures with new data. The data look better, i.e. narrower linewidth, more apparent derivative-like features. The authors also rewrote the manuscript in the attempt to make a more convincing case about the novelty of the work. I must say the writing is improved, and my comments are addressed quite satisfactorily. There is still one argument that I disagree. For example, in the rebuttal, the authors tried to compare the optical Stark effect (OSE) works between those in valley TMDs and GaAs QWs, and mentioned that a clear polarization-selective OSE has been done years ago in GaAs system. This is not quite the case however. In TMDs, the two valleys can be controlled separately by two circularly polarized light because they are independent due to the finite and opposite Berry curvatures. In GaAs, there will always be mixtures of states being induced with circularly polarized light because GaAs system is more like a mixture of s- and p-like orbitals in atomic systems. That is, these states are very close to each other in energy, and also not quite a clean system for such experiments, i.e. regardless of the pump polarization, there will always be a finite OSE for a given probe polarization, even theoretically.

I am still fixed to my previous standpoint regarding the novelty. However, I must admit that the revised manuscript is much more improved in terms of data quality and writing. I do foresee that this work can be of significant interests to other researchers in the field. Therefore, I think this work can be suitable for a publication in Nature Communications.

Response:

We appreciate the time that Reviewer 1 took to read revised manuscript and responses to reviewers. We are quite pleased with Reviewer 3's comment that our work can be suitable for a publication in Nature Communications. Regarding the selectivity of optical Stark effect in existing materials, we completely agree with Reviewer 3 that monolayer group VI TMDs have superior property compared to GaAs-based systems. As Reviewer 1 pointed out, while perfect selectivity of the optical Stark effect is not allowed in conventional semiconductors due to unavoidable mixture of orbitals, the non-trivial Berry phases make it possible to independently control the valley states in monolayer group-VI TMDs. For this reason, we have revised a sentence in the manuscript, as below. We thank again Reviewer 3 for his/her thoughtful comment.

- Original sentence (line 56, page 3)

It has recently been discovered that the optical Stark shift of excitons in different valleys in momentum space can be determined by changing helicity of light in monolayer WSe₂, a group VI TMD^{13,14}.

- Revised sentence (line 57, page 3)

It has recently been discovered that the optical Stark shift of excitons in different valleys in momentum space can be determined by changing helicity of light in monolayer group VI TMDs (WS₂ and WSe₂) in a completely selective manner^{13,14}.